# Equilibrium Studies on Pd(II)–Amine Complexes with Bio-Relevant Ligands in Reference to Their Antitumor Activity

**DOI:** 10.3390/ijms24054843

**Published:** 2023-03-02

**Authors:** Mohamed M. Shoukry, Rudi van Eldik

**Affiliations:** 1Department of Chemistry, Faculty of Science, University of Cairo, Giza 12613, Egypt; 2Department of Chemistry and Pharmacy, University of Erlangen-Nuremberg, 91058 Erlangen, Germany; 3Faculty of Chemistry, Nicolaus Copernicus University in Torun, 87-100 Torun, Poland

**Keywords:** Pd-amine complexes, speciation studies, bio-relevant ligands, stability constants, thermodynamic parameters

## Abstract

This review article presents an overview of the equilibrium studies on Pd-amine complexes with bio-relevant ligands in reference to their antitumor activity. Pd(II) complexes with amines of different functional groups, were synthesized and characterized in many studies. The complex formation equilibria of Pd(amine)^2+^ complexes with amino acids, peptides, dicarboxylic acids and DNA constituents, were extensively investigated. Such systems may be considered as one of the models for the possible reactions occurring with antitumor drugs in biological systems. The stability of the formed complexes depends on the structural parameters of the amines and the bio-relevant ligands. The evaluated speciation curves can help to provide a pictorial presentation of the reactions in solutions of different pH values. The stability data of complexes with sulfur donor ligands compared with those of DNA constituents, can reveal information regarding the deactivation caused by sulfur donors. The formation equilibria of binuclear complexes of Pd(II) with DNA constituents was investigated to support the biological significance of this class of complexes. Most of the Pd(amine)^2+^ complexes investigated were studied in a low dielectric constant medium, resembling that of a biological medium. Investigations of the thermodynamic parameters reveal that the formation of the Pd(amine)^2+^ complex species is exothermic.

## 1. Introduction

The reactions of transition metal ions have received much attention due to their importance in biology and medicine. Some metal ion complexes have been used for the treatment of several diseases such as cancer, arthritis, diabetes, Alzheimer’s, etc. [1]. Bioinorganic chemistry is developing a better understanding of the mechanism of action of metal complexes in biological systems [2,3]. 

The application of metal complexes in medicine received much attention after the discovery of the most widely used antitumor drug, cisplatin, in 1967 by Rosenberg [4]. Research in this area led to the synthesis of many chemotherapeutic agents. Only a few of them have entered clinical application, most are still subject to preclinical investigations [5,6]. 

The development of new platinum compounds is now in progress with the aim being to find new drugs that: have fewer side effects, can overcome drug resistance during therapy, and afford a broader range of applications [7,8,9,10,11]. The platinum complexes developed include N-donor, containing monodentate, bidentate or tridentate ligands [12,13,14,15,16,17]. Furthermore, Pt(II) complexes with bidentate N,S ligands were found to be promising cytostatic agents [18,19,20,21], for example dichloro(2-methylthiomethylpyridine)platinum(II).

Farrell’s group developed a new class of binuclear Pt(II) complexes with improved properties in terms of toxicity or cross-resistance to cisplatin [22,23,24,25,26,27]. The apparent advantage of these complexes is the high charge (+4), compared to neutral mononuclear complexes. It resulted in good solubility, efficient electrostatic interaction with anionic DNA (the major target for the drug) and rapid uptake. The homo- and hetero-binuclear complexes of Pd^2+^ and/or Pt^2+^ were investigated [28]. Complexes of the type [Pd_2_(tpbd)Cl_2_]Cl, [Pt_2_(tpbd)Cl_2_]Cl and [PdPt(tpbd)Cl_2_]Cl were synthesized and characterized, where tpbd = N,N,N’,N’’-tetrakis(2-pyridylmethylbenzene-1,4-diamine). The Pt(II) complex showed the most powerful cytotoxic effect on HTB140 and H460 cancer cell lines.

Great interest in the investigation of Pd(II) complexes [29] with amines possessing various functional groups began with the discovery of cisplatin and its antitumor activity. Various simple Pd(II) complexes have interesting biological properties, as illustrated in Figure 1.

The amines investigated are di- and tridentate aliphatic and aromatic amines possessing extra functional groups, namely thioether and alcoholato groups. The structure of the amine has an effect on the reactivity of the Pd(II) complex. Both Pd(II) and Pt(II) complexes have a similar general structure and thermodynamic properties. The complexes formed with Pd(II) are, however, five orders of magnitude more labile than their platinum counterparts [28]. Palladium analogues are investigated instead of, or as well as, the platinum (II) complexes. It is proposed that the faster aquation of Pd(II) than Pt(II) in vivo makes the former a better model for studies on the interaction with bio-ligands existing in biological fluids. The present article presents equilibrium data on various Pd-amine complexes. The effect of the structural parameters of the amines on the complex formation equilibria are investigated. The interaction of these complexes with DNA, the major target in tumor therapy, together with amino acids, peptides and dicarboxylic acids, are reviewed. The effect of solvent polarity and thermodynamic parameters is contributing to the significance of these complexes. Also, the speciation of binuclear Pd(II)-amine complexes is of great importance, based on Farrell’s group research results. 

## 2. Crystal Structure of Pd(II) Complex with Bidentate Amine

The Pd(II) complexes with amines possessing different functional groups were synthesized and characterized in many studies [30,31,32,33,34,35,36]. The crystal structure of the Pd(II) complex with bidentate amine 1,4-bis(2-hydroxylethyl)piperazine (BHEP) [30], was investigated. The palladium center shows a square-planar geometry with a tetrahedral distortion. The two chloride ions in the coordination environment occupy the cis position, and the ligand (BHEP) is bound by piperazine nitrogen atoms to form a five-membered metallocyclic ring, with a distorted twist boat conformation. The Pd–Cl bond lengths (2.309(3) and 2.306(3) Å) can also be regarded as normal in comparison to distances found in the literature (2.220–2.361) [37]. However, the Cl–Pd–Cl angle shows a small deviation from the square-planar geometry and shows a value of 90.57(3)°.

## 3. Acid–Base Equilibria of Pd-Amine Complex

### 3.1. Acid–Base Equilibria of Pd-Bidentate Amine Complexes [Pd(N^N)(H_2_O)_2_]^2+^

The acid–base equilibria and complex formation equilibria with Pd-bidentate species were investigated using [Pd(N^N)(H_2_O)_2_]^2+^ instead of [Pd(N^N)Cl_2_]. This may be due to the high stability of Pd-Cl bonds [38]. Therefore, the stability constant of the Pd(N^N)-ligand system refers to the equilibrium between [Pd(N^N)(H_2_O)_2_]^2+^ and the ligand, and not between [Pd(N^N)Cl_2_] and the ligand. For this reason, the chloride ion in [Pd(N^N)Cl_2_] should be replaced by a weakly coordinating anion, either nitrate or perchlorate. The coordinated water molecules in [Pd(N^N)(H_2_O)_2_]^2+^ are usually more acidic than bulk water molecules. The acid–base chemistry was characterized by fitting the potentiometric data to different acid–base models. The best-fit model was found to be consistent with three species, 10-1, 10-2 and 20-2. These numbers are the stoichiometric coefficients of a complex with a general formula [Pd(N^N)(H_2_O)_2_]_l_ L_p_ H_q_. The first two species are due to the deprotonation of the two coordinating water molecules, as given in Equations (1) and (2). The third species, 20-2, is the dimeric di-µ-hydroxo complex of two 10-1 species, according to Equation (3). The dimeric species were also evidenced in the case of dibromo(ethylenediamine) using electrospray mass spectrometry [39].
[Pd(N^N)(H_2_O)_2_]^2+^
  $\overset{pK_{a1}}{⇌}$
 [Pd(N^N)(H_2_O)(OH)]^+^ + H^+^      (1)10010-1
[Pd(N^N)(H_2_O)(OH)]^+^
  $\overset{pK_{a2}}{⇌}$
 [Pd(N^N)(OH)_2_] + H^+^       (2)10-110-2

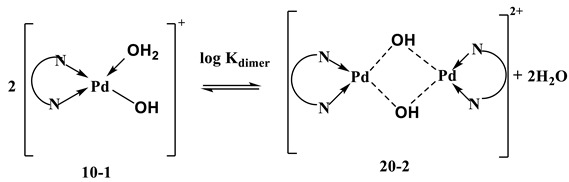
(3)

The equilibrium constant for the dimerization reaction, Equation (3), can be calculated using Equation (4).
log K_dim_ = log β_20-2_ − 2 log β_10-1_(4)

In the case of Pd(II) complexes with bidentate amine ligands in the presence of bulky substituents on the nitrogen donor atoms, in regard to the Pd(II) complex with 2-{[2-dimethylamino)-ethyl]methylamino} ethanol [31], the formation of the dihydroxo-bridged dimer (20-2) was not detected. This may be due to the bulky substituents that undergo steric interaction, making the formation of the dihydroxo-bridged dimer (20-2) energetically unstable. The µ-hydroxo species (20-1) are formed through the dimerization of the Pd(II) complex via a hydroxo-group, as in Equation (5).

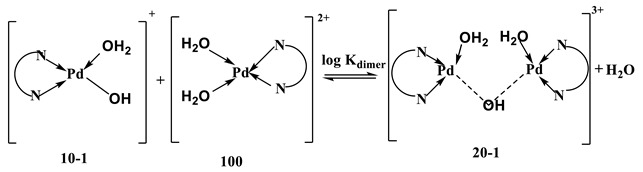
(5)

The equilibrium constant for the dimerization reaction (4) can be calculated using Equation (6).
log K_dim_ = log β_20-1_ − log β_10-1_(6)

The hydrolysis of [Pd(N^N)(H_2_O)_2_]^2+^ was evidenced by spectral measurements. The spectral band [30] of [Pd(BHEP)(H_2_O)_2_]^2+^ occurring at 360 nm undergoes a blue shift to 345 nm upon formation of [Pd(BHEP)(H_2_O)OH]^+^ (10-1 species), with the addition of one equivalent of NaOH. This is due to the ligand field splitting as a result of the deprotonation of a coordinating water molecule. This band is further shifted to 335 nm with the addition of two equivalents of NaOH forming the dihydroxo species, [Pd(BHEP)(OH)_2_] (10-2 species).

Changes from N,N- to N,S-chelated Pd(II) are accompanied by a change in the acid–base equilibria of the complex, as in the [Pd(MME)(H_2_O)_2_]^2+^ complex [32], where MME is methionine methyl ester, and in the [Pd(SMC)(H_2_O)_2_]^+^ complex, where SMC is S-methyl-L-cysteinate [33]. The hydrolysis does not lead to the formation of the dihydroxo–bridged dimer (20-2). The µ-hydroxo species (20-1) are formed as explained previously [33]. This may be accounted for on the basis that the strong labilization effects of the S-donor atom may cause the formation of the cycle in the 20-2 species not to be favored, and consequently the dimeric form (20-2) will be strained and not favored energetically. Both the pK_a1_ and pK_a2_ values of the water molecules coordinated to the Pd-amine complexes depend on the nature of the N donors, see Table 1. The pK_a1_ value for the picolylamine (pic) complex [34], is intermediate between those of the [Pd(en)(H_2_O)_2_]^2+^ [35] and [Pd(bpy)(H_2_O)_2_]^2+^ [36] complexes. This is explained on the premise that picolylamine has one pyridine ring that has π-acceptor properties, leading to an increase in the electrophilicity of the Pd ion, and consequently a decrease in the pK_a_ of the coordinated water molecules. However, the [Pd(bpy)(H_2_O)_2_]^2+^ complex has two pyridine rings, see Figure 1. 

### 3.2. Acid–Base Equilibria of Pd-Tridentate Amine Complexes, [Pd(N^N^N)(H_2_O)]^2+^

The acid–base chemistry was studied by fitting the potentiometric data for the [Pd(N^N^N)(H_2_O)]^2+^ complex to different acid–base forms. The best fit was found to be consistent with the species of stoichiometric coefficients 10-1 and 20-1, see Equations (7) and (8). The dimer (20-1) with a single hydroxo-bridge can be formed by the reaction shown in Equation (8), as reported before [40]. The equilibrium constant of the dimerization reaction is shown in Equation (9).

[Pd(N^N^N)(H_2_O)]^2+^

  $\overset{pK_{a1}}{⇌}$
 [Pd(N^N^N)(OH)]^+^ + H^+^      (7)10010-1


(8)
[Pd(N^N^N)(H2O)]2++[Pd(N^N^N)(OH)]+ ⇌logKdimer[(N^N^N)Pd(µ-OH)Pd(N^N^N)]3++H2O


log K_dim_ = log β_20-1_ − log β_10-1_(9)

## 4. Interaction of [Pd(N^N)(H_2_O)_2_]^2+^ Complexes with Amino Acids

The Pd(N^N)-amino acid complexes are reported to have better antitumor activity than the cisplatin. These complexes [41] have shown growth inhibition against L1210 Lymphoid leukemia, P388 Lymphocytic leukemia and Sarcama180. Some of them show I.D_50_ values lower than cisplatin. The structure of these complexes was investigated by using NMR measurements. Also, the speciation of the complexes with amino acids possessing different functional groups, were reported.

The interaction of the [Pd(en)(H_2_O)_2_]^2+^ complex with amino acids was investigated by ^15^N-NMR spectral measurements [42]. Glycine forms [Pd(en)(gly-N-O)]^+^ were in the pH range 4-10. Above pH 10, [Pd(en)(gly-N)(OH)] is formed. At high pH and with excess glycine, [Pd(en)(gly-N))_2_] predominates. At low pH, near pH 2, [Pd(en)(gly-O)(H_2_O)_2_]^2+^ exists in equilibrium with the N,O-chelate complex, free glycine and [Pd(en)(H_2_O)_2_]^2+^ species. The reactions of [Pd(en)(H_2_O)_2_]^2+^ with β-alanine were nearly similar to those of glycine, except for the N,O-chelate complex that was less stable. However, the reaction with γ-aminobutyric acid in the pH range 4-8, yielded the chelate complex [Pd(en)(γ-aba-N,O)]^+^ with the species [{Pd(en)(µ-γ-aba)}]^2+^. The reaction of [Pt(en)(H_2_O)_2_]^2+^ with glycine was investigated by ^15^N-NMR in a solution of pH 1.7. Moreover, [Pt(en)(Hgly-O)](H_2_O)_2_]^2+^ was formed after 2 weeks, and converted completely to [Pt(en)(gly-N,O)]^+^. 

It has been well established that N,O-chelation is a characteristic coordination mode for glycinate bound to Pd(II). The [Pd(N^N)(H_2_O)_2_]^2+^ complexes with amino acids were extensively investigated by using potentiometric measurements. The composition and stability of the formed complexes were estimated and the concentration distribution diagrams were evaluated. The amino acids with no functional groups in the side chains generally form 1:1 complexes with [Pd(N^N)(H_2_O)_2_]^2+^. Their complexes are very stable, with stability constants of logβ_110_ ~ 8–12 [43]. They coordinate through both the amino acid group and carboxylate oxygen, forming stable five-membered chelate rings. The effect of chelate ring size and the basicity of the amino acid group on the stability of the formed complexes were studied [44]. The stability constant logβ_110_ of [Pd(AEMP)-α-alanine]^2+^ (9.74) is larger than that of [Pd(AEMP)-β-alanine]^2+^ (9.10), (AEMP = 2-(2-aminoethyl)-1-methylpyrrolidine). This is the result of the extra stabilization of the 5-membered chelate rings for the α-alanine complex, as compared to the 6-membered rings for the β-alanine complex [44]. The same trend was found for the Pd(dmen) system (dmen = N,N-dimethylethylenediamine) [45]. The proline complex has the highest stability constant among various amino acids. This may be a result of the higher basicity of the proline amino group, as reflected by its higher pK_a_ value [46]. Concentration distribution diagrams for the complexes [Pd(BHEP)(H_2_O)_2_]^2+^ with amino acids [30], show the formation of 110 species at low pH and predominates in the physiological pH range with a maximum concentration of 83%. The monohydroxo species, 10-1, do not contribute significantly (lower than 20%), whereas the dihydroxo species only start to form at a higher pH.

The reactions of [Pd(N^N)(H_2_O)_2_]^2+^ complexes with amino acids were investigated by using spectral measurements. The electronic spectra [30] of [Pd(BHEP)(H_2_O)_2_]^2+^ shows a band at 360 nm. This band is shifted to 312 nm for [Pd(BHEP)(glycine)]^+^ (110 species). This shift is expected as a result of the ligand field splitting upon substitution of the coordinated water by glycine. 

Serine and threonine are α-amino acids possessing β-OH groups in the side chain. They have two dissociable protons in the measurable pH range (-NH_3_^+^ and -COOH). The alcoholic hydroxy group does not deprotonate in the measurable pH range (pKa > 14). The participation of the alcoholic OH group in complex formation reactions, was reported [47,48,49]. The [Pd(N^N)(H_2_O)_2_]^2+^ interacts with selected amino acids, forming complex species of stoichiometric coefficients, namely 110 and 11-1. The formation of species 11-1, shows the ionization of the alcoholic OH group. The pK_a_ of the ionization was determined (pK_a_ = logβ_110_–logβ_11-1_) as 9.60 and 9.38 for serine and threonine complexes, respectively, with [Pd(AEMP)(H_2_O)_2_]^2+^ [44]. The corresponding value for the serine complex with [Pd(pic)(H_2_O)_2_]^2+^ (where pic is 2-picolylamine) is 8.30 [34]. Lowering the pK_a_, indicated stronger acidification of the OH group in the [Pd(pic)(H_2_O)_2_]^2+^ complex. This may be due to the π-accepting property of the picolylamine pyridine group, which will increase the electrophilicity of the Pd (II) ion, and result in a decrease in the pK_a_ value of the coordinated OH group. Ethanolamine forms a complex [44] with [Pd(AEMP)(H_2_O)_2_]^2+^ with a lower stability constant, at logβ_110_ = 7.93. This is accounted for on the basis that there is no neutralization of the charges involved during the complex formation with ethanolamine. The pK_a_ of ionization of the coordinated alcoholic OH group (pK_a_ = 6.81) is more acidic than those of serine and threonine. The lower pK_a_ value of the ethanolamine complex can be accounted for on the basis that the stronger coordination of the hydroxyl oxygen atom to the palladium center lowers the pK_a_ of the OH group. The OH bond is weakened, with subsequent deprotonation occurring at lower pH values. Ethanolamine forms in addition to the 110 and 11-1, and the 120 species. The latter species was not formed in the case of serine and threonine, respectively. 

The concentration distribution diagram for the [Pd(AEMP)(H_2_O)_2_]^2+^ complex with threonine^42^ is shown in Figure 2. This figure indicates that threonine forms the species 110 at low pH, dominates between pH 4.5–8.0, and thus prevents the hydrolysis of Pd(II), i.e., the species (10-1) is either not present or present at very low concentration. The species (10-2) is only present at pH > 10. The ionization of the OH group (11-1) starts around pH~7.0 and dominates at pH~10.2

Protonated histidine contains three dissociable protons, which can dissociate in the following sequence: carboxylic acid, imidazolium N(3)-H^+^ and the side chain ammonium NH_3_^+^ group. The imidazole N(1)-H is weakly acidic (pK_a_ = 14.4), and does not dissociate in the measurable pH range [50]. The donor atom preferences in the coordination of histidine were studied by using NMR measurements [51]. The reaction of cis-[Pt(NH_3_)_2_(H_2_O)_2_]^2+^ with histidine in an acid medium of pH 2-3, leads to binding of the carboxylate oxygen. This is followed by ring closure, during which the amino group nitrogen and carboxylate oxygen are coordinated. This complex remains stable up to pH 8–9. Increasing the pH, leads to deprotonation of the imidazole nitrogen, which is followed by coordination of the amino group nitrogen and imidazole nitrogen atoms. In the reaction with the [Pd(N^N)(H_2_O)_2_]^2+^ species, only two of the above three coordination sites are involved in the complex formation reaction. The stability constant of the histidine complex is in fair agreement with that of histamine and is higher than those of other amino acids. This reveals that histidine is coordinating via the amino and imidazole nitrogen atoms, in the same way as histamine does. Concentration distribution diagrams on histidine and histamine complexes show that the 110 species is the main species in the physiological pH range. Thus, it is assumed that the coordination of histidine and histamine prevents the hydrolysis of Pd(II), so the species (10-1) and (10-2) are either not present or present at very low concentration. The species (10-2) is only present at pH > 11.

Ornithine and lysine are α-amino acids possessing an extra terminal amino group. They coordinate to [Pd(N^N)(H_2_O)_2_]^2+^ as bidentate chelate, either with two amino groups (N,N) or glycine-like, through the α-amino and carboxylate groups (N,O). The results show that both kinds of chelation are involved in different proportions, depending on the pH of solution and the chelate ring size. 

Ornithine forms a more stable complex than α-amino acids. This may be taken as evidence that ornithine most likely coordinates via the two amino groups at a higher pH. This is supported by the strong affinity of palladium to nitrogen donor centers. However, lysine forms a complex with a stability constant in fair agreement with those of α-amino acids. This could indicate that lysine most likely chelates with the α-amino and carboxylate groups (glycine-like). A complex formed through binding with the two amino groups will lead to an unstable eight-membered ring.

At low pH, ornithine and lysine chelate via carboxylate and one amino group, leaving the other amino group to get protonated and form the protonated 111 species. The pK_a_ values of the protonated complexes given by (pK_a_ = logβ_111_ − logβ_110_) with [Pd(AEMP)(H_2_O)_2_]^2+^ [44], are 8.79 and 10.54 for ornithine and lysine, respectively. The pK_a_ of the ornithine complex is lower than that of lysine, which may reveal that at a higher pH of the terminal amino acid group is most likely involved in coordination in the case of ornithine (N,N-coordination), and less likely involved in the case of lysine (glycine-like). The speciation diagrams show that ornithine and lysine start to form the protonated species 111 at low pH and dominate in the physiological pH range. The 110 species is predominantly formed at pH > 10.5 [44]. 

Amino acids containing sulfur, e.g., methionine and S-methyl cysteine, react very rapidly with Pd(II) and Pt(II) because of the great tendency of sulfur (soft Lewis base) to bind with these metals (soft Lewis acids). Methionine and S-methyl cysteine have three coordination sites. The coordination site preference in binding was elucidated by using NMR measurements. The reaction of cis-[Pt(NH_3_)_2_(H_2_O)_2_]^2+^ with methionine and S-methyl cysteine in an acidic medium involves the formation of an O,S chelate complex. The complex is subsequently isomerized to the N,S-chelate complex. The conversion from O,S- to N,S-chelate complex is irreversible. The coordination of Pt(II) and Pd(II) with methionine in an aqueous solution reveals coordination through the sulfur atom, and chelation through the amino group in a second, slow step [52,53]. 

The stability constants of methionine and S-methyl cysteine complexes with [Pd(N^N)(H_2_O)_2_]^2+^ are lower than those of simple amino acids [44]. This may be accounted for on the basis that the amino acid group of methionine and S-methyl cysteine, is less basic than those of other amino acids, as reflected by their pK_a_ values. The concentration distribution diagrams of methionine and S-methylcysteine complexes [44] show that the 110 species dominate up to pH~11.

Glutamic acid is a α–amino acid with a γ-carboxylic acid group in the side chain. It has three functional groups that undergo acid–base reactions. The three dissociable protons in the measured pH range are those corresponding to NH_3_^+^, α-COOH and γ-COOH groups. In coordination with [Pd(N^N)(H_2_O)_2_]^2+^, only two of the above three binding sites are involved during complex formation. The complexes detected are the 111 and 110 species. The stability constants of Pd(N^N)-glutamate (110), are in fair agreement with those of α-amino acids. Thus, coordination is assumed to occur via the α-amino acid and carboxylate groups. The concentration distribution diagram 42 indicates that the protonated complex 111 predominates at low pH, and the 110 species predominates in the pH range 4 to 11. 

## 5. Interaction of [Pd(N^N)(H_2_O)_2_] ^2+^ with Peptides

Amide bonds or groups provide the linkage between adjacent amino acids. A protein is composed of a chain of (n) amino acids and contains (n-1) peptide (amide) bonds in the backbone. The tetrahedral amino nitrogen in an amino acid with pK_a_ ~ 9.7, loses its basicity upon reaction to give trigonal nitrogen in an amide bond. Amide groups, Figure 2, are planar due to a 40% double-bond character in the C-N bond, and the *trans* form is strongly favored [54]. 

The amide group is a very weak acid for proton loss from the trigonal nitrogen to give a negative species. For acetamide, the pK_a_ value is reported to be 15.1, and 14.1 for glycylglycinate. The weak acidity makes quantitative equilibrium measurements very difficult.

For simple dipeptides, there are at least four donor groups (amino-N, carboxylate-O, amide-N and carbonyl-oxygen) of which all are capable of metal ion coordination. Terminal amino and carboxylate groups are the most effective binding sites for metal ions, because of the neutrality of the amide group. Under the influence of metal ions, such as Pd(II), the amide group coordinates through induced ionization of the peptide hydrogen.

[Pd(N^N)(H_2_O)_2_]^2+^ + L^+^
 ⇌ [Pd(N^N)L]^+^ + 2 H_2_O      (10)
(110)

[Pd(N^N)L]^+^
 ⇌ [Pd(N^N)LH_-1_] + H^+^           (11)(110)(11-1)

The potentiometric data of the Pd(N^N)-peptide system is best fitted considering the formation of the species 110 and 11-1, according to Figure 3, and Equations (10) and (11). The species 110 is formed by binding the peptide amino and carbonyl groups, see Equation (10). Upon increasing pH, the amide group is ionized and the coordination sites switch from carbonyl oxygen to amide nitrogen, leading to the formation of species 11-1, see Equation (11). This switching of coordination sites was reported for peptide complexes [55,56,57]. Glutamine forms a more stable complex than glycinamide. This is due to the fact that glycinamide is neutral and glutaminate is a mono-negatively charged ion. The pK_a_ values of the coordinated amide group in the Pd(II) complex (log β_110_ − log β_11-1_) are in the range of 4.19 to 9.78. These values are higher than those for Pd(Pic)-peptide complexes [34]. This is a result of the π-accepting property of the picolylamine pyridine group. 

It is interesting to compare the pK_a_ values of the peptide complexes. The pK_a_ value for the glycinamide complex has the lowest value. This is discussed on the basis that the more bulky substituent on the peptide hinders the switching of the binding sites from carbonyl oxygen to amide nitrogen. The speciation diagram of the Pd(AEMP)-glycylglycine system is shown in Figure 3. The 110 species, start to form at low pH and the concentration increases with increasing pH, attaining a maximum concentration of 82% at pH 5.5. A further increase of pH is accompanied by decreasing the 110 species concentration and increasing the 11-1 species concentration, [Pd(AEMP)LH_-1_]^+^, which is the main species at pH > 7.6. Therefore, the 11-1 species predominates in the physiological pH range.

The electronic absorption spectra of Pd(AEMP)-glycylglycine complexes were investigated. Spectral bands of [Pd(AEMP)(H_2_O)_2_]^2+^ and its glycylglcine (Glygly) complex are quite different in the position of the maximum wavelength, see Figure 4. The spectral band of the [Pd(AEMP)(H_2_O)_2_]^2+^ complex (A) showed up at 360 nm, and is shifted to 325 nm upon formation of the [Pd(AEMP)(Glygly)]^+^ complex (B). A further shift of the band to 313 nm appeared under deprotonation and formation of the [Pd(AEMP)(Glygly-H)] complex (C). The progressive shift toward a shorter wavelength in the absorption spectrum, may indicate evidence for the induced ionization of the amide proton upon complex formation. This is supported by the potentiometric measurements for the coordination scheme of peptides. There is no significant absorption for glycylglycine in this spectral region.

The DFT calculations were carried out to investigate the equilibrium geometry of the peptide ligands and their complex species with Pd(N^N)(H_2_O)_2_^2+^. The optimized structures of the 110 and 11-1 species are stable configurations as evidenced by potentiometric measurements. The palladium center has a typical square-planar geometry with some distortion. Ligands act as bidentate chelates, forming a five-membered ring. The values of angles around the Pd atom are near to 90°. The more negative the value of total energy of the complex [Pd(AEMP)GlyGly]^+^ (−1004.17 a.u.) compared to that of [Pd(AEMP)GlyA]^2+^ (−776.64 a.u.) indicates that the former is more stable. This is in accordance with the potentiometric data that a stronger coordination of glycylglycine (log β_110_ = 7.48) compared to glycinamide (log β_110_ = 7.32) occurs. 

The potentiometric results indicated that the coordinated peptide NH is more acidic in case of glycinamide (pK_a_ = 4.19) than glycylglycine (pK_a_ = 6.99) for Pd(AEMP)- peptide complex. This is in accordance with the DFT calculation, as the calculated formal charge on the hydrogen atom of the coordinated peptide NH group in glycinamide (+0.397) is more positive than that of glycylglycine (+0.309). Also, the calculated formal charge set on the peptide carbonyl carbon atom is more positive in the coordinated glycinamide (+0.456) than that of glycylglycine (+0.406). The more positive charge will enhance the release of the peptide hydrogen, and consequently decrease the pK_a._ [44]. 

## 6. Interaction of [Pd(N^N)(H_2_O)_2_]^2+^ Complexes with Dicarboxylic Acids

Among the Pt(II) complexes that have received world-wide approval, is carboplatin. It has cyclobutane dicarboxylic acid coordinated to the Pt(II) center. The antitumor activity of cisplatin and carboplatin were compared. The species distribution of both species is identical, in the low concentration range on the micro-molar scale. However, at a higher concentration on the milli-molar scale, their concentration distribution differs. At high concentration, cisplatin has a higher percentage of the hydroxo species than carboplatin, especially inside the cell. These data, in addition to some kinetic reasons, explain the results that the nephrotoxicity of cisplatin is higher than that of carboplatin [58,59,60,61]. For this particular reason, it was interesting to study the dicarboxylic acid complexes with the [Pd(N^N)(H_2_O)_2_]^2+^ species. 

The acid–base equilibria of cyclobutane-1,1-dicarboxylic acid were further investigated. The pK_a1_ and pK_a2_ values are 3.00 and 5.57, respectively, at 25 °C and 0.10 M ionic strength [62]. The interaction of cyclobutane dicarboxylic acid with [Pd(N^N)(H_2_O)_2_]^2+^ species leads to the formation of the 110 and 111 species, as shown in Figure 4.

The pK_a_ values of the protonated species for the [Pd(N^N)HL]^+^ are lower than those of HL^−^, where HL^−^ represents the dicarboxylate ion possessing one proton. This indicates acidification of the second carboxylic acid group upon coordination of the Pd to one carboxylate group. The pK_a_ value of this protonated species in the case of [Pd(en)(HCBDCA)]^+^ was estimated before using UV-Vis measurements to be ca. 2.5 at 25 °C and 0.10 M ionic strength [63]. 

The chelate ring size plays a role on the stability of the dicarboxylic acid complexes [62,64]. The formation constants of the 1:1 complexes possessing five- and six-membered chelate rings, as in cyclobutane dicarboxylic acid, oxalic acid and malonic acid, are higher than those involving a seven-membered chelate ring, as in succinic acid. This maybe based on the fact that the five- and six-membered rings are more favored energetically than the seven-membered ring. It is interesting to note that the CBDCA has a higher stability constant than malonic acid, although both of them form six-membered chelate rings. This may be due to the higher pK_a_ values of the former than the latter, dicarboxylic acid. Succinic acid forms the protonated complex (111) in addition to the complex species (110). 

The concentration distribution diagram for Pd(N^N)-CBDCA complexes [62], shows that the protonated species (111) predominate only at low pH. The species (110) are present as the main species in the physiological pH range. The mono-coordinated species was documented to be the active form in the case of carboplatin [65]. However, the displacement of chelated CBDCA by DNA (as 5’-GMP) cannot be ruled out [66]. Also, it is interesting to note that the hydrolyzed species does not exist, which may be taken as a reason why carboplatin is more active than cisplatin, where the hydrolyzed species predominates in case of cisplatin. The non-formation of the hydrolyzed species in carboplatin is due to the fact that the coordinated CBDCA protects it from the side reactions like those with a OH^−^ and chloride ion. 

### Interaction of [Pd(N^N)(CBDCA)] with DNA

Cisplatin is known to have low solubility and affinity to hydrolyze in neutral media. These are the disadvantages of cisplatin. The solubility in water is increased and the toxicity, and as a result the hydrolysis is reduced, by displacement of the chloride ions by the carboxylate groups, as in CBDCA. Carboplatin was suggested to be a pro-drug for cisplatin [64]. It was reported that the antitumor activity is due to a ring-opening reaction of carboplatin, followed by the interaction with guanosine 5-monophosphate to form a quaternary complex [67] [Pt(NH_3_)_2_(CBDCA-O)(5-GMP)]. The kinetics of the ring-opening reaction by DNA constituents, was studied in the reaction of [Pd(amine)(CBDCA)] with inosine 5-monophosphate [63]. 

The ring-opening reaction was investigated from a thermodynamic point of view. The complex formation equilibria in the system containing [Pd(Pic)(H_2_O)_2_]^2+^, CBDCA and DNA constituents [34], such as uracil, uridine, thymine, IMP or GMP, were investigated using the potentiometric technique. The quaternary complex of the general formula (Pd(Pic))_l_(CBDCA-O)_p_(DNA)_q_(H)_r_, with stoichiometric coefficients l, p, q and r, is formed according to Equation (12).
(12)l Pd(Pic) + p (CBDCA-O) + q DNA ↔ (Pd(Pic))l(CBDCA-O)p(DNA)q(H)r.

The proposed structure of the quaternary complex with IMP is shown in Figure 5.

The potentiometric data were fitted considering different models of different composition. For pyrimidines, the accepted model is consistent with the formation of the 1110 species. The nucleotides IMP and GMP form the 1110 and 1111 species. The pK_a_ values of the protonated complexes (1111) are 6.21 for IMP and 6.51 for GMP, respectively. These most probably correspond to the protonated phosphate groups. It is interesting to find that the quaternary complexes of pyrimidines are less stable than those of the nucleotides IMP and GMP. This may be discussed on the premise that the cyclobutane ring forms a close hydrophobic contact with the purine rings of IMP and GMP. Such hydrophobic contact may enhance the stabilization of the quaternary complexes. The same result was obtained from an NMR investigation of the carboplatin–GMP complex [67]. 

The speciation diagram obtained for the Pd(Pic)–CBDCA–GMP system is shown in Figure 6. The Pd(Pic)–CBDCA species (1100) predominates with a maximum concentration of 54% at pH = 4.0. The Pd(Pic)–GMP species (1010) reaches the highest concentration of 18% at pH = 8.2. The quaternary species Pd(Pic)–CBDCA–IMP (1110) attains a maximum degree of formation of 80% in the pH range 8.6 to 9.0. This reveals that the ring-opening of chelated CBDCA by DNA is possible in the physiological pH range.

## 7. Effect of Chloride Ion Concentration on the Activity of Antitumor Complexes

Antitumor Pt(II)-amine complexes, are usually administrated in the chlorido form. Under the high chloride ion concentration (of 0.10 M) that exists in human blood plasma, this form predominates. The chlorido complex is electrically neutral. This property facilitates its passage to the inside of the cell, where the chloride ion concentration is much lower (ca. 4 mM). Under these conditions the chlorido complex is subjected to aquation and hydrolysis. The effect of chloride ion concentration on the complex formation equilibria of Pd(II)-amine complexes with biorelevant ligands, is considered as a realistic extrapolation to biologically relevant conditions. The activity of the CBDCA toward different Pd(II) species increases markedly when the chloride ion concentration decreases in the reaction medium. This is in accordance with the conclusion reached by Lim and Martin in the case of [Pt(en)Cl**_2_**], based on the equilibrium distribution of Pt(II)(en) and the rates of reactions of pyridine with Pt(II)(dien) complexes [68]. 

The chloride ion concentration in biological media has a significant effect on the stability constants of the Pd(pic)-CBDCA complex. The effect of chloride ion concentration on the stability of the CBDCA complex [34] was investigated at a constant ionic strength of 0.30 M. The stability constant of the Pd(pic)-CBDCA complex decreases upon increasing the [Cl^−^]. This may be explained on the premise that the increase of [Cl^−^] is accompanied by a decrease in the concentration of the active mono- and diaqua complexes, which is accompanied by a decrease in the stability constant of the complex. 

## 8. Interaction of [Pd(N^N)(H_2_O)_2_]^2+^ with DNA Constituents

It was reported that DNA is the major target of antitumor complexes. The mode of action of the antitumor activity is well established. The complexes are assumed to substitute their chloride ions (leaving groups) to coordinate with the suitably oriented nitrogen base units on DNA [69,70,71,72]. Therefore, the formation equilibria of [Pd(N^N)(H_2_O)_2_]^2+^ complexes with DNA, are very significant from the biological point of view, and supports the antitumor activity of the drugs. 

The complexes of DNA constituents with [Pd(N^N)(H_2_O)_2_]^2+^ [73] are formed and predominate with the formation factor, depending on the pH of the medium. Pyrimidines, such as uracil, thymine and thymidine, have basic nitrogen donor atoms (N3) in the measurable pH range [74,75], with pK_a_ > 9. As a result, complex formation predominates above pH 8.5 to form 1:1 and 1:2 complexes with [Pd(N^N)(H_2_O)_2_]^2+^ species. The thymine complex was found to be more stable than that of uracil. This is accounted for based on the higher basicity of the N3 site of thymine than uracil. This most probably results from the inductive effect of the extra electron-donating methyl group in thymine. Inosine and nucleotides, such as inosine-5′-monophosphate and adenosine-5′-monophosphate, are coordinated [73] giving the 1:1 complex and its protonated form, in addition to the 1:2 complex. The protonated inosine complex undergoes proton ionization for which a pK_a_ value of 4.80 was determined. This value corresponds to N1H and is lowered with respect to that of free inosine (pK_a_ = 8.80). The lowering of the pK_a_ is due to acidification upon complex formation [76]. The IMP complex is more stable than that of inosine. This can be discussed on the basis of different coulombic forces that operate between the ions resulting from the negatively charged phosphate group. Interaction between the phosphate group and exocyclic amine through hydrogen, is assumed to contribute to the increased stability. Such interaction through hydrogen bonding was reported previously for related systems [77,78]. 

Deactivation of the Antitumor by S-Ligands and Amino Acids 

Nitrogen-donor ligands, such as DNA constituents and nucleosides, have a high affinity to [Pd(N^N)(H_2_O)_2_]^2+^. This may have important biological significance since the reaction with DNA is thought to be responsible for the antitumor activity of related complexes [79,80,81,82]. However, the high preference of Pd(II) to coordinate to S-donor ligands, cannot be ignored. This suggests that Pd(II)-N adducts can easily be converted into Pd-S adducts [82,83]. Consequently, the equilibrium constant for such substitution reactions is of biological significance. Consider inosine as a typical DNA constituent (presented by HA) and cysteine as a typical thiol ligand (presented by H_2_B). The equilibria involved in complex formation with [Pd(N^N)(H_2_O)_2_]^2+^ [45] and the displacement reactions are given below, Equations (13)–(17).

[Pd(N^N)(A)]^+^ + B^2−^

  $\overset{K_{\mathrm{eq}}}{⇌}$
 [Pd(N^N)(B)] + A^−^
      (13)

The equilibrium constant for the displacement reaction given in Equation (14) is given by
K_eq_ = [Pd(N^N)(B)][A^−^]/[Pd(N^N)(A)^+^ ][B^2−^](14)
[Pd(N^N)]^2+^ + A^−^

 ⇌ [Pd(N^N)A]^+^              (15a)(100)(110)
β_110_ [Pd(N-N)A]^+^ = [Pd (N^N)A^+^]/[Pd(N^N)^2+^][A^−^](15b)
[Pd(N^N)]^2+^ + B^2−^
 ⇌ [Pd(N^N)B]               (16a)(100)(110)
β_110_[Pd(N-N)B] = [Pd(N^N)B]/[Pd(N^N)^2+^][B^2−^](16b)

Substitution from Equations (14b) and (15b) in Equation (16) results in:K_eq_ = β_110_^[Pd(N^N)B]^/β_110_^[Pd(N^N)A]+^(17)

The equilibria and displacement reactions for Pd(dmen)A complex were investigated [45]. The potentiometric data of the inosine complex was fitted by considering the formation of the 1:1 complex and its protonated form, in addition to the 1:2 complex. The log β values are 6.51 ± 0.04, 10.48 ± 0.04 and 11.19 ± 0.05 for the 110, 120 and 111 species, respectively. Cysteine forms a 1:1 complex and its protonated form. The log β values are 16.33 ± 0.03 and 20.59 ± 0.03 for the 110 and 111 species, respectively. The log β_110_ values for [Pd(dmen)(A)]^+^ and [Pd(dmen)B] complexes amount to 6.51 and 16.33, respectively, and by substitution in Equation (17) results in log K_eq_ = 9.72 [45]. In the same way the equilibrium constants for the displacement of coordinated inosine by glycine and methionine are log K_eq_ = 3.74 and 2.91, respectively. These values clearly indicate how sulfide ligands [83,84], such as cysteine and glutathione, are effective in displacing the DNA constituent, i.e., the main target in tumor chemotherapy.

## 9. Complex Formation Equilibria of Binuclear Pd(ΙΙ) Complexes and Some Selected DNA Constituents

Farrell set a milestone in developing new non-classical platinum compounds, in fact more precisely, polynuclear Pt(II) complexes, for example the trinuclear BBR3464 or the binuclear system [85,86,87,88]. The binuclear Pt(II) complexes are considered to be third generation antitumor complexes [89,90]. In these complexes the two platinum centers are linked through a flexible bridge, such as an aliphatic chain [91] or azole molecules forming a rigid bridge [92]. Some of these binuclear complexes are now subject to clinical tests. The reason for increasing the antitumor activity in binuclear complexes, is their ability to bind to DNA better than cisplatin and related complexes [93]. The reported results led to a completely different antitumor activity. Earlier studies showed that the binuclear Pt(II) complexes using azole as a linker, have better antitumor activity in vitro in cisplatin resistance cell lines, as well as in several antitumor cell lines [94,95]. The high activity of this class of complexes is described on the basis of the structural features, viz. the azole-bridged complexes possess a leaving hydroxo group, an appropriate Pt—Pt distance and some flexibility to provide the 1,2-interstrand cross-links with a minimal distortion of DNA. Most of the binuclear complexes were investigated in the solid phase. Solution equilibria of these systems support their biological activity. Their speciation provides information regarding to what extent these complexes are formed in biological media. Also, it shows the feasibility of the interaction of the binuclear complex with DNA, the major target in tumor therapy. The linker used in our studies is 1,4-bipiperidine. 

The complex formation equilibria of the binuclear complexes are proposed to occur according to Figure 5. The results of the equilibrium data for 1,4-bipiperidine complex with [Pd(BHEP)(H_2_O)_2_]^2+^ [30], taken as an example, reveal the formation of the complex species 111, 110 and 210, with formation constant values of 13.72, 19.91 and 21.05, respectively. The coefficients correspond to [Pd(BHEP)(H_2_O)_2_]^2+^, 1,4-bipiperidine and proton, respectively. 

The concentration distribution diagram for the Pd(BHEP)-1,4-bipiperidine system [30] reveals that the binuclear complex (210) reaches a maximum formation degree of 94.4% in the pH range between 3.2 and 8.6. The complete formation of the 210 complex species is indicated by the sharp inflection at the equivalence point, with an overall formation constant of log β_210_ = 21.05. After this point, the binuclear complex is subjected to hydrolysis. The hydrolysis equilibria are given in Figure 6. Analysis of the potentiometric data supports the formation of the species 10-1 and 10-2, with formation constants of –6.67 and –13.83, respectively. The coefficients correspond to [(H_2_O)(BHEP)Pd(Bip)-Pd(BHEP)(H_2_O)]^4+^, ligand and proton, respectively. Speciation data reveals that the 10-1 species start to form at pH 4.0 and reach the maximum concentration of 48% at pH 6.8. The 10-2 species start to form at pH 5.8 and its concentration increases with increasing pH.

The formation of the binuclear complex is further evidenced by spectral measurements [30]. The results show a band at 361 nm for the [Pd(BHEP)(H_2_O)_2_]^2+^ species. This band is shifted to 341 nm for the [(H_2_O)(BHEP)Pd-Bip-Pd(BHEP)(H_2_O)]^4+^ species. This band is further shifted to 333 nm through the addition of an excess of NaOH for the [(OH)(BHEP)Pd-bip-Pd(BHEP)(OH)]^2+^ species to form. It should also be noted that there is no UV absorption for bipiperidine in the spectral measurement range.

The interaction of the [(H_2_O)(N^N)Pd-(Bip)-Pd(N^N)(H_2_O)]^4+^ species with DNA constituents, such as inosine, uracil and uridine, leads to the formation of 110 and 120 complexes, as described in Figure 7. The formation constant values for the BHEP system are 6.62, 10.55 (for inosine), 7.16, 12.12 (for uracil) and 7.01, 11.95 (for uridine), respectively. 

The speciation diagram of the [(H_2_O)(BHEP)Pd-Bip-Pd(BHEP)(H_2_O)]^4+^- Uracil system, Figure 7, reveals that the uracil complexes predominate in the physiological pH range. Therefore, the interaction with DNA is quite feasible. This supports the antitumor activity of the binuclear complex.

## 10. Thermodynamic Parameters of [Pd(N^N)(H_2_O)_2_]^2+^ Complexes

Currently, the majority of the thermodynamic results on the coordination of [Pd(N**^**N)(H_2_O)_2_]^2+^ complexes were estimated at 25 °C. Consequently, further investigations will continue to estimate the thermodynamic data at higher temperatures, including formation constants, enthalpy and entropy terms. Characterization of the thermodynamic parameters of a carboplatin model, as with the Pd(N^N)(CBDCA) complex, is of biological significance as it helps to understand the chemical behavior of antitumor complexes. The hydrolysis constants of [Pd(N^N)(H_2_O)_2_]^2+^, the protonation constants of CBDCA and the formation constants of the complexes with CBDCA, were calculated at different temperatures and a 0.10 M ionic strength. The thermodynamic parameters ΔH and ΔS were obtained by using the linear least squares fit of lnK versus 1/T, leading to an intercept of ΔS/R and a slope of -ΔH/R [96]. The results for [Pd(BHEP)(H_2_O)_2_]^2+^ complexes were reported [64]. The main conclusions obtained are summarized below.

(i) Hydrolysis of [Pd(N^N)(H_2_O)_2_]^2+^

The hydrolysis reactions are accompanied by endothermic liberation of the ordered water of hydration from the reactants to form bulk water of greater disorder, and are accompanied by a significant increase in entropy. However, the ΔH values can be considered as the net sum of two opposing effects, namely the exothermic hydrolysis reaction and the endothermic liberation of ordered water of hydration. The hydrolysis reactions corresponding to the deprotonation of coordinated water molecules are exothermic reactions. The first hydrolysis step comprises attraction between the doubly positive ion and OH^−^, whereas the second hydrolysis step comprises attraction between the mono-cation and OH^−^. Thus, the first reaction is more exothermic than the second one. Also, the first water molecule to be removed in the first reaction is stronger bound than the second water molecule in the second reaction, and results in the ΔS of the first reaction being larger than that of the second one. 

The dimerization is an exothermic process, but has a negative entropy change despite the release of two water molecules. This is attributed to the rearrangement due to the dimerization reaction, which contributes to a negative entropy (higher order).

(ii) Formation of Pd(N^N)(CBDCA) complex

The reactions leading to the protonated and deprotonated forms are exothermic reactions with negative ΔH values. This is attributed to charge neutralization between the positively [Pd(N^N)(H_2_O)_2_]^2+^ and negatively charged cyclobutanedicarboxylate ions. 

## 11. Solvent Effect

Biological media are of a lipophilic character. It is well established that the “effective” or “equivalent solution” dielectric constants in proteins or active site cavities of enzymes [97,98,99,100,101], are smaller compared to those in bulk water. Estimates for the dielectric constants in such locations range from 30 to 70. Consequently, the biological medium is better represented by a non-aqueous medium. Thus, by using aqueous solutions containing ≈10–50% dioxane, one may expect to simulate the situation to some degree in a biological environment. Currently, most of the thermodynamic data on the [Pd(N^N)(H_2_O)_2_]^2+^ complexes were estimated using aqueous media. This may be the result of the belief that “in vivo” media are represented by aqueous media. Estimation of the thermodynamic data in dioxane-water mixtures for a carboplatin model, such as the [Pd(N^N)(CBDCA)] complex, supports the biological activity of this class of coordination compounds. Investigation of the effect of the solvent dielectric constant of the medium on the complex formation equilibria of [Pd(N^N)(CBDCA)], requires data on the acid–base equilibria of CBDCA and the hydrolysis of [Pd(N^N)(H_2_O)_2_]^2+^ in the same solvent.

The [Pd(N^N)(H_2_O)_2_]^2+^ is subjected to hydrolysis by deprotonation of the coordinated water molecules. There is an increase in pK_a1_ and pK_a2_ upon increasing the organic solvent content. This is explained as follows, by increasing the organic solvent percentage leads to a decrease in the dielectric constant of the medium. This leads to an increase in the pK_a_ of coordinated water molecules, which is in accordance with the electrostatic model [102,103,104,105].

The [Pd(N^N)(H_2_O)_2_]^2+^ forms with CBDCA two complex species, the protonated form [Pd(N^N)(HCBDCA)]^+^ and the deprotonated form [Pd(N^N)(CBDCA)]. Careful examination of the medium effect on the equilibrium constants of the [Pd(N^N)CBDCA] system [34], reveal the following features:The formation constants of the protonated complex [Pd(N^N)HCBDCA] increases with the addition of dioxane to an aqueous solution of the corresponding species. This can be explained as a result of increasing the electrostatic forces between [Pd(N^N)(H_2_O)_2_]^2+^ and the mono-negatively charged HCBDA ion, leading to extra complex stability. The monoprotonated complex undergoes proton ionization, the pK_a_ for this ionization step increases with the increase of dioxane content, as described previously.The stability constant of the [Pd(N^N)(H_2_O)_2_]^2+^ complex with cyclobutane dicarboxylic acid increases upon the addition of dioxane. This is due to the increased electrostatic attraction between the doubly negatively charged cyclobutane dicarboxylate ion and [Pd(N^N)(H_2_O)_2_]^2+^.

## 12. Conclusions

The Pd(II) complexes are more reactive than their Pt(II) counterparts. For this reason, the Pd(II) complexes are good models for the analogous Pt(II) complexes in solution. Equilibrium data for the Pd(II)-amine complexes with ligands commonly existing in biological fluids, support the antitumor activity of the Pt(II) complexes. The formation equilibria of the Pd-amine complexes with bio-relevant ligands, such as amino acids, peptides, DNA constituents and dicarboxylic acids, were discussed. Using the combination of stability constant data of these complexes, it will be possible to estimate the equilibrium distribution of the antitumor drug in biological fluids where all types of ligands are present. This would form a clear basis for understanding the behavior of antitumor drugs under physiological conditions. From the equilibrium data of the Pd(II) complexes of various amines possessing different functional groups, it is possible to design some amines with structural parameters that enhance the reaction with DNA. This will lead to more drugs having better antitumor activity.

Amino acids form highly stable complexes. The substituent on the α-carbon atom of the amino acid has a significant effect on the stability of the formed complex. The substituent having an extra amino group, as in ornithine and lysine or the imidazole group as in histidine, increase the stability constant due to the high affinity of Pd(II) to the nitrogen donor group. The substituent possessing a carboxylic acid group, as in aspartic acid and glutamic acid, does not contribute to the stability of the formed complex. This may be explained on the premise that the additional carboxylic acid group is not competing with the amino group during complex formation. A substituent with the alcoholic OH group plays a role in complex formation. The Pd(II) complex promotes the ionization of the OH group through complex formation.

The pyrimidine constituents of DNA, such as uracil, uridine and thymine, form 110 and 120 complexes with [Pd(N^N)(H_2_O)_2_]^2+^. They act as monodentate ligands, and as a result of their high pK_a_ values (pK_a_ ≈ 9) the complexes start to form above pH 6. The binding centers are the negatively charged nitrogen sites, and the complexes predominate in the neutral and slightly basic medium. Purines, such as inosine and guanosine, form the 110 and 120 species. The 111 species is formed in acidic solution and corresponds to the N_7_ coordinated complex, while N_1_ nitrogen is in its protonated form. The purine nucleotides, such as inosine-5′-monophosphate and guanosine-5′-monophosphate, form the 110, 111 and 112 complexes. The protonated complexes are formed in the acidic pH range and correspond to the N_7_ coordinated species. Their complexes are more stable than those of pyrimidines. The extra stability is due to the different Coulombic forces operating between the ions resulting from the negatively charged phosphate groups. Also, possible hydrogen bonding involving the phosphate group may contribute to the high stability of the nucleotide complexes.

It was reported that the antitumor activity of carboplatin is due to the ring-opening of chelated CBDCA and the interaction with DNA. Therefore, the complex formation equilibria of [Pd(N^N)(H_2_O)_2_]^2+^ with dicarboxylic acids involving CBDCA is of interest. Also, the equilibria involved in the ring-opening and the reaction with DNA constituents forming quaternary complexes, is very important from a biological point of view, and the results are expected to support the antitumor activity of carboplatin. A lower polarity has been detected in some biochemical micro-environments, such as active sites of enzymes and the side chain of proteins. These properties approximately correspond to those existing in dioxane/water mixtures. For these reasons, the formation equilibria of the [Pd(N^N)(CBDCA)] complex, taken as a representative example, in dioxane-water solutions of different compositions could be of biological significance.

## Data Availability

Not applicable.

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
