# Peer review of "Equilibrium Studies on Pd(II)–Amine Complexes with Bio-Relevant Ligands in Reference to Their Antitumor Activity"

_ijms, 2023, doi:10.3390/ijms24054843_

Round 1

Reviewer 1 Report

In this manuscript entitled “Equilibrium studies on Pd(II)–amine complexes with bio-rele-vant ligands in reference to their antitumor activity” Eldik and co-workers summarized the equilibrium studies of antitumor active  Pd(II)–amine complexes.

The equilibrium studies play a crucial role to determine the stability parameters of the metal complexes and are also helpful to tune the structures of the ligands. I think this manuscript will provide a brief overview of the complex’s stability and is greatly helpful for the readers, with this I recommend this manuscript for publication after major revision. To improve the quality of the manuscript authors could address the below comments.  

 1.       Typo error in line 5 in the abstract. Similarly, several formatting and typo issues in the manuscript.

2.   Authors could provide the compound numbers. 

3.       Instead of platinum complexes in scheme 1. I strongly encourage providing the palladium complexes with antitumor activity or other biologically active complexes.

4.       The majority of the introduction is general and there is no specific discussion of the objective of the manuscript. Instead, of focusing on Pt-based metal complexes. The author could provide the importance of this topic and a literature overview of the chosen topic and cite the recent reviews in the introduction.

5.       The authors could provide the importance of the equilibrium studies of Pd(II)–amine complexes in the introduction and their future perspective in the conclusion.

6.       Authors could provide the reference for the below sentences “The Pd(II) complexes with amines having different functional groups were synthesized and characterized in many studies”

7.       It will be helpful to the readers. if the authors provide the timeline of the reports. 

Author Response

Reviewer 1

  1. Typo error in line 5 in the abstract. Similarly, several formatting and typo issues in the manuscript.

We think the error in line 5 is the missed “.”  It is now given in red.

  1.  Authors could provide the compound numbers.

Most of the compound numbers are now included in Scheme 1 

  1. Instead of platinum complexes in scheme 1. I strongly encourage providing the palladium complexes with antitumor activity or other biologically active complexes.

A list of some palladium(II) complexes with biological activity, scheme 1, page 4

  1. The majority of the introduction is general and there is no specific discussion of the objective of the manuscript. Instead, of focusing on Pt-based metal complexes. The author could provide the importance of this topic and a literature overview of the chosen topic and cite the recent reviews in the introduction.

The importance of the investigated topic and the discussion of the objective was given in page 4 (red)

  1. The authors could provide the importance of the equilibrium studies of Pd(II)–amine complexes in the introduction and their future perspective in the conclusion.

The importance of the equilibrium studies of Pd(II)-amine complexes was given in page 4 (red).

The future perspective was given in the conclusion, lines 12-15 (red)

  1. Authors could provide the reference for the below sentences “The Pd(II) complexes with amines having different functional groups were synthesized and characterized in many studies”

References are now given (30-34).

  1. It will be helpful to the readers if the authors provide the timeline of the reports.

This is easier said than done! This would mean that the complete manuscript has to be rewritten?

Reviewer 2 Report

The review proposed by Shoukry and Eldik for publication in the international Journal of Molecular Sciences entitled «Equilibrium studies on Pd(II)–amine complexes with bio-relevant ligands in reference to their antitumor activity» bring relevant information about the behavior of Pd(II)-amine complexes into biological systems. 

  • In the abstract, I disagree with the sentence “Such systems are considered as good models for all possible reactions occurring with antitumor drugs in biological systems” because the biological system is much more complex than the simulated conditions of the study and is a bit challenging to simulate ALL possible reactions.
  • In the introduction, in the first paragraph “The mechanism of action in biological systems2,3 are not well understood (…)”, I think this generalization isn’t correct since we have the light of many possible action mechanisms of many compounds including cisplatin. 
  • In the introduction, the first sentence of the last paragraph, “Pd(II) and Pt(II) complexes have the same general structure and thermodynamic properties”. In this sentence, I think the word SIMILAR is more appropriate than the word SAME. 
  • Furthermore, I missed the introduction of more information about Pd complexes with antitumoral activities. There are many of them described in the literature. 
  • In my opinion, a detailed description of the crystal structure of Pd(II) complex with bidentate amine isn’t necessary for this article since the author already described them in another work. 
  • In section 3 the sentence “For this reason, the chloride ion in [Pd(N^N)Cl2] should be replaced by a non-coordinating anion as nitrate or perchlorate, that do not form stable bonds with Pd(N^N)2+” is incorrect. Nitrate and perchlorate could coordinate with metal ions in monodentate or chelating modes, but these ions are weakly coordinated with metal ions. I think is necessary to change the sentence. 
  • I missed a Figure with the structures described in table 1. In section 4, the sentence “Pd(N^N)-amino acid complexes are reported to have better antitumor activity than the cis-platin, as they have fewer side effects.38” is incorrect! In the cited article the Pd(N^N)-amino acid complexes are much less active in tumor cells than cisplatin.
  • Fig.7 is from ref 42 is necessary to add this information in the legend. 
  • Throughout the text, I missed the representation of many structures that are discussed. For instance, Pd(Pic)–CBDCA–IMP. 
  • Almost all Figures in this article are from other references. I think many of them aren’t necessary.
  • The conclusion is so extensive, it seems to be an overview of the article. 

Author Response

Reviewer 2

  • In the abstract, I disagree with the sentence “Such systems are considered as good models for all possible reactions occurring with antitumor drugs in biological systems” because the biological system is much more complex than the simulated conditions of the study and is a bit challenging to simulate ALL possible reactions.
  • This sentence was changed to “……Such systems may be considered as one of the models for the possible reactions occurring with antitumor drugs in biological system.

  • In the introduction, in the first paragraph “The mechanism of action in biological systems2,3 are not well understood (…)”, I think this generalization isn’t correct since we have the light of many possible action mechanisms of many compounds including cisplatin. 

This paragraph was changed to Bioinorganic chemistry is growing to have a better understanding of the mechanism of action of metal complexes in biological systems.2,3

  • In the introduction, the first sentence of the last paragraph, “Pd(II) and Pt(II) complexes have the same general structure and thermodynamic properties”. In this sentence, I think the word SIMILAR is more appropriate than the word SAME. 

The word same was substituted by similar

  • Furthermore, I missed the introduction of more information about Pd complexes with antitumoral activities. There are many of them described in the literature. 

In the introduction, information about Pd(II) complexes with anti-tumor activity was cited in Ref. 29.

  • In my opinion, a detailed description of the crystal structure of Pd(II) complex with bidentate amine isn’t necessary for this article since the author already described them in another work.

The description of the crystal structure was summarized.

  • In section 3 the sentence “For this reason, the chloride ion in [Pd(N^N)Cl2] should be replaced by a non-coordinating anion as nitrate or perchlorate, that do not form stable bonds with Pd(N^N)2+” is incorrect. Nitrate and perchlorate could coordinate with metal ions in monodentate or chelating modes, but these ions are weakly coordinated with metal ions. I think is necessary to change the sentence. 

The sentence was changed to For this reason, the chloride ion in [Pd(N^N)Cl2] should be replaced by a weakly coordinated anion as nitrate or perchlorate.”

  • I missed a Figure with the structures described in table 1. In section 4, the sentence “Pd(N^N)-amino acid complexes are reported to have better antitumor activity than the cis-platin, as they have fewer side effects.38” is incorrect! In the cited article the Pd(N^N)-amino acid complexes are much less active in tumor cells than cisplatin.

Figure with the structures described in table 1 was given, page 8.

In section 4, the sentence “Pd(N^N)-amino acids…..” was changed to These complexes39 have shown growth inhibition against L1210 Lymphoid leukemia , P388 Lymphocytic leukemia and Sarcama180. Some of them show I.D.50 values lower than cis-platin.” With new references.

  • Fig. 7 is from ref 42 is necessary to add this information in the legend. 

The information is added (from Ref. 42)

  • Throughout the text, I missed the representation of many structures that are discussed. For instance, Pd(Pic)–CBDCA–IMP. 

The structure of Pd(Pic)-CBDCA-IMP was given in Fig. 8 on page 22.

  • Almost all Figures in this article are from other references. I think many of them aren’t necessary.

Some Figures were deleted

  • The conclusion is so extensive, it seems to be an overview of the article. 

We have shortened the conclusions section

Reviewer 3 Report

In this manuscript, entitled “Equilibrium studies on Pd(II)–amine complexes with bio-relevant ligands in reference to their antitumor activity” the authors discuss the equilibria of Pd-amine complexes with ligands such as amino acids, dicarboxylic acids, peptides, and DNA constituents. Since the behaviour of Pd(II) complexes resembles that of their Pt(II) analogues in solution, the equilibrium data of Pd-amine complexes might help to elucidate the antitumour activity of Pt(II) drugs. Overall, this paper is very informative and well-conceived. However, some issues could be addressed in order to improve the readability of the manuscript.

The major issues include:

-          Given the high number of paragraphs and sub-paragraphs, I believe that writing a summary at the beginning of the paper might improve the readability. For the same reason, describing the paper’s structure at the end of the introduction might enhance the overall comprehension;

-          In some parts of the paper, it seems that some references are missing. Some examples are:

o   In paragraph 5 “This is in accordance with the potentiometric data that a stronger coordination of glycylglycine (logβ110 = 7.48) compared to glycinamide (logβ110 = 7.32) occurs.”;

o   Again, in paragraph 5 “The more positive charge will enhance the release of the peptide hydrogen and consequently decrease the pKa”;

o   In paragraph 6.1 “Cis-platin is known to have low solubility and affinity to hydrolyze in neutral media. These are the most disadvantages of cis-platin. The solubility in water is increased and the toxicity as a result of the hydrolysis is reduced, by displacement of chloride ions by carboxylate groups as in CBDCA. Carboplatin was suggested to be a pro-drug for cisplatin.”;

-          For such a massive review, it seems that there are few references. Moreover, many of them seem to be outdated. If possible, please add more recent references;

-          In paragraph 4, I think it would improve the readability if it was specified at the beginning how the sub-paragraphs are organized. Moreover, in order to help readers to better understand the topic, if possible, I would expand the introduction as made in the other paragraphs (i.e. paragraph 7);

Minor issues of readability include:

-          Some of the structures reported in scheme 1 seem to have some bonds thicker than the others. Please, replace the figure with another of better quality. Moreover, I think that the drug name “cisplatin” should be written always in the same way. In the introduction it is reported as “cis-platin”, whereas in scheme 1 as “cisplatin”. Furthermore, in the rest of the text, sometimes it has been written in italics;

-          In table 1, some of the numbers are not correctly aligned;

-          Some of the equations (like number 2 on page 4) seem not to be well aligned;

-          Some sections of the text seem to be written with different fonts or formatting, like paragraphs 4.1 and 8;

-          Some sentences seem to be not very clear.

o   For example, in paragraph 3.1 “This may be accounted for on the basis that the strong labilization effects of the S-donor atom will cause the dimeric form (20-2) to be strained and consequently, and not favored energetically”;

o   In paragraph 6 “The non-formation of the hydrolyzed species in carboplatin is that the coordinated CBDCA protects it from the side reactions as those with OH- and chloride ion”;

-          Sometimes it seems that some concepts are repeated. For example, in paragraph 6: “Although, both CBDCA and malonic acid form six-membered chelate rings, the CBDCA complex has a higher stability constant. This may be explained on the premise that CBDCA has higher pKa values than malonic acid. It is interesting to note that CBDCA has a higher stability constant than malonic acid, although both of them form six-membered chelate rings. This may be due to the higher pKa values of the former than the latter dicarboxylic acid.”;

-          Sometimes it is difficult to find the equations to which the species are referred. Specifying it in brackets might improve the readability;

-          If possible, please report the reaction scheme for the ring-opening reaction of carboplatin presented in paragraph 6.1;

In conclusion, I believe that this paper is of high importance and definitely worthy of publication. Regarding the implementations I suggested, I think they would be beneficial for the enhancement of the understanding of the paper. 

Author Response

Reviewer 3

The major issues include:

-          Given the high number of paragraphs and sub-paragraphs, I believe that writing a summary at the beginning of the paper might improve the readability. For the same reason, describing the paper’s structure at the end of the introduction might enhance the overall comprehension;

In the introduction, a summary describing the paper’s structure was given in red, page 3 (bottom)

-          In some parts of the paper, it seems that some references are missing. Some examples are:

o   In paragraph 5 “This is in accordance with the potentiometric data that a stronger coordination of glycylglycine (logβ110 = 7.48) compared to glycinamide (logβ110 = 7.32) occurs.”;

The more negative value of total energy of the complex [Pd(AEMP)GlyGly]+ (-1004.17 a.u.) compared to that of [Pd(AEMP)GlyA]2+ (-776.64 a.u.) indicates that the former is more stable. This is in accordance with the potentiometric data that a stronger coordination of glycylglycine (logβ110 = 7.48) compared to glycinamide (logβ110 = 7.32) occurs.  The energy calculation results are in agreement with the stability constant results.

  • Again, in paragraph 5 “The more positive charge will enhance the release of the peptide hydrogen and consequently decrease the pKa”;

Ref. 42 was given in red.

  • In paragraph 6.1 “Cis-platin is known to have low solubility and affinity to hydrolyze in neutral media. These are the most disadvantages of cis-platin. The solubility in water is increased and the toxicity as a result of the hydrolysis is reduced, by displacement of chloride ions by carboxylate groups as in CBDCA. Carboplatin was suggested to be a pro-drug for cisplatin.”;

Ref. 64 was given in red.

-          For such a massive review, it seems that there are few references. Moreover, many of them seem to be outdated. If possible, please add more recent references;

Some recent references were added

-          In paragraph 4, I think it would improve the readability if it was specified at the beginning how the sub-paragraphs are organized. Moreover, in order to help readers to better understand the topic, if possible, I would expand the introduction as made in the other paragraphs (i.e. paragraph 7);

The paragraph 4 was improved by adding these sentences

The structure of these complexes was investigated by NMR measurements. Also, the speciation of the complexes with amino acids having different functional groups, were reported.

Minor issues of readability include:

-          Some of the structures reported in scheme 1 seem to have some bonds thicker than the others. Please, replace the figure with another of better quality. Moreover, I think that the drug name “cisplatin” should be written always in the same way. In the introduction it is reported as “cis-platin”, whereas in scheme 1 as “cisplatin”. Furthermore, in the rest of the text, sometimes it has been written in italics;

Scheme 1 was deleted and replaced by another one.

-          In table 1, some of the numbers are not correctly aligned;

Table 1 was corrected

-          Some of the equations (like number 2 on page 4) seem not to be well aligned;

The equations were aligned

-          Some sections of the text seem to be written with different fonts or formatting, like paragraphs 4.1 and 8;

Paragraphs 4.1 and 8 were written with the same font.

-          Some sentences seem to be not very clear.

  • For example, in paragraph 3.1 “This may be accounted for on the basis that the strong labilization effects of the S-donor atom will cause the dimeric form (20-2) to be strained and consequently, and not favored energetically”;

This paragraph was rewritten as This may be accounted for on the basis that the strong labilization effect of the S-donor atom may cause the formation of the cycle in 20-2 species not favored and consequently the dimeric form (20-2) will be strained and not favored energetically.”

  • In paragraph 6 “The non-formation of the hydrolyzed species in carboplatin is that the coordinated CBDCA protects it from the side reactions as those with OH- and chloride ion”;

Cis-platin is aquated. The coordinated water molecules can be substituted by OH-and Cl-ions. However, in case of carboplatin, there is no coordinated water molecules.

-          Sometimes it seems that some concepts are repeated. For example, in paragraph 6: “Although, both CBDCA and malonic acid form six-membered chelate rings, the CBDCA complex has a higher stability constant. This may be explained on the premise that CBDCA has higher pKa values than malonic acid. It is interesting to note that CBDCA has a higher stability constant than malonic acid, although both of them form six-membered chelate rings. This may be due to the higher pKa values of the former than the latter dicarboxylic acid.”;

The repeated sentences were deleted.

-          Sometimes it is difficult to find the equations to which the species are referred. Specifying it in brackets might improve the readability;

Equation numbers were given in round brackets

-          If possible, please report the reaction scheme for the ring-opening reaction of carboplatin presented in paragraph 6.1;

The reaction scheme was given as:

The quaternary complex of general formula (Pd(Pic))l(CBDCA-O)p(DNA)q(H)r, with stoichiometric coefficients l, p, q and r is formed according to Eq. (11)

l Pd(Pic) + p (CBDCA-O) + q DNA    (Pd(Pic))l(CBDCA-O)p(DNA)q(H)r    (11)

The proposed structure of the quaternary complex with IMP is given in Fig. 5.

Round 2

Reviewer 1 Report

In the updated manuscript authors responded to all questions and included suggestions. I recommend this manuscript to publish in IJMS.

Reviewer 3 Report

All the issues raised have been properly addressed. Now, the manuscript can be accepted for publication in IJMS.